# Design of a Capacitorless DRAM Based on a Polycrystalline-Silicon Dual-Gate MOSFET with a Fin-Shaped Structure

**DOI:** 10.3390/nano12193526

**Published:** 2022-10-09

**Authors:** Hee Dae An, Sang Ho Lee, Jin Park, So Ra Min, Geon Uk Kim, Young Jun Yoon, Jae Hwa Seo, Min Su Cho, Jaewon Jang, Jin-Hyuk Bae, Sin-Hyung Lee, In Man Kang

**Affiliations:** 1School of Electronic and Electrical Engineering, Kyungpook National University, Daegu 41566, Korea; 2Korea Multi-Purpose Accelerator Complex, Korea Atomic Energy Research Institute, Gyeongju 38180, Korea; 3Power Semiconductor Research Center, Korea Electrotechnology Research Institute, Changwon 51543, Korea; 4DB HiTek, RF/Mixed Signal Development Team, Eumseong 27605, Korea

**Keywords:** dual-gate, grain boundary, polycrystalline silicon, 1T-DRAM, metal-oxide-semiconductor field-effect transistor, sensing margin, retention time

## Abstract

In this study, a capacitorless one-transistor dynamic random-access memory (1T-DRAM) cell based on a polycrystalline silicon dual-gate metal-oxide-semiconductor field-effect transistor with a fin-shaped structure was optimized and analyzed using technology computer-aided design simulation. The proposed 1T-DRAM demonstrated improved memory characteristics owing to the adoption of the fin-shaped structure on the side of gate 2. This was because the holes generated during the program operation were collected on the side of gate 2, allowing an expansion of the area where the holes were stored using the fin-shaped structure. Therefore, compared with other previously reported 1T-DRAM structures, the fin-shaped structure has a relatively high retention time due to the increased hole storage area. The proposed 1T-DRAM cell exhibited a sensing margin of 2.51 μA/μm and retention time of 598 ms at *T* = 358 K. The proposed 1T-DRAM has high retention time and chip density, so there is a possibility that it will replace DRAM installed in various applications such as PCs, mobile phones, and servers in the future.

## 1. Introduction

Dynamic random access memory (DRAM) has been employed as a memory element for decades and is one of the key components of electronic devices [1,2]. This is because hat conventional DRAM has one transistor–one capacitance, simple configuration, high chip density, and low power consumption. In recent years, miniaturization of electronic devices has necessitated the use of small cells; however, it is extremely difficult to reduce the size of capacitors. To address this concern, various structures such as 3D storage capacitors, cylindrical vertical array transistors, and trench cell capacitors have been proposed [3,4,5]. However, such structures complicate manufacturing processes. Therefore, a one-transistor (1T)-DRAM that can achieve a high chip density by eliminating the need for a capacitor, which is the basic hurdle in reducing the size of a DRAM cell, has been proposed. However, planar 1T-DRAM still exhibits poor retention time. Therefore, various structures and methods for improving the memory characteristic have been investigated [6,7,8,9,10,11,12,13,14,15,16,17].

The primary concept of 1T-DRAM involves the implementation of memory characteristics using floating body effects with a silicon-on-insulator (SOI) structure [18,19,20]. However, the SOI technology is expensive, and its mass production is limited. Nevertheless, cost issues can be overcome by using a polycrystalline silicon (poly-Si)-based SOI-like structure. In addition, poly-Si-based 1T-DRAM can be combined with the technologies used in 3D NAND manufacturing to increase chip density.

In this paper, we proposed a novel 1T-DRAM based on a poly-Si dual-gate metal-oxide-semiconductor field-effect transistor (MOSFET) with a fin-shaped structure. The proposed device adopts a fin-shaped structure to improve the memory characteristics and can be 3D-stacked using poly-Si, which can increase the chip density. During program operation, holes are generated by band-to-band tunneling (BTBT) between gates 1 and 2 in the body region, and these holes are stored on the side of gate 2. Therefore, a fin-shaped structure is adopted on the side of gate 2 to expand the storage space and improve the retention time. In addition, the advantage of the fin-shaped structure is that the retention time can be improved without changing the size of the device. Furthermore, we optimize the storage region length, storage region height, and body thickness, which are important geometric parameters for memory characteristics, and achieve high retention times. Furthermore, to improve the accuracy of the simulation, a study is conducted assuming the existence of a single grain boundary (GB) at the center of the body region.

## 2. Device Structure and Simulation Method

Figure 1a shows a cross-sectional view of the proposed poly-Si dual-gate MOSFET-based 1T-DRAM with a fin-shaped structure. Gate 1 was used for the conventional MOSFET and programming operations. Gate 2 was used to perform the program, erase, and hold operations. The work functions of gate 1 (WF_G1_) and gate 2 (WF_G2_) are 4.85 eV and 5.3 eV, respectively. Note that poly-Si and Ni were employed as electrode materials for gates 1 and 2, respectively [21,22]. The proposed device was designed with a fin-shaped structure to increase the storage area where holes accumulated and to improve the retention time. Figure 1b is a simple circuit diagram of 1T-DRAM, and the capacitor is removed from the DRAM circuit diagram.

Figure 2 shows the key fabrication steps of the proposed 1T-DRAM based on a poly-Si dual-gate MOSFET with a fin-shaped structure crystallized via excimer laser crystallization consisting of a total of 11 steps [23]. First, metal is deposited on an oxidized silicon wafer, and then dry etching is performed to form a bottom gate electrode. Second, after depositing SiO_2_ serving as a spacer, dry etching of the SiO_2_ in the bottom gate electrode area is carried out. Third, HfO_2_, the gate dielectric, is deposited and etched. Fourth, a-Si is deposited using LPCVD. Fifth, excimer laser irradiation is performed to convert a-Si to poly-Si. Sixth, etching of poly-Si is performed to make a fin shape. Seventh, HfO_2_ and metal are deposited and etched to form the top gate electrode. Eighth, ion implantation is performed to form the n-type source and drain. Ninth, SiO_2_ is deposited for insulation between the gate electrode and the source and drain electrodes. Tenth, the source and drain metal is deposited. Finally, after removing SiO_2_ from the top of the electrode, SiO_2_ is deposited for passivation [23].

The gate length (*L*_g_) and gate dielectric (HfO_2_) thickness (*T*_ox_) were 100 and 3 nm, respectively. The doping concentrations in the source, body, and drain regions were 1 × 10^20^ (*n*-type), 1 × 10^18^ (*p*-type), and 1 × 10^20^ (*n*-type), respectively. The relevant parameters of the proposed device are presented in Table 1. Note that in the proposed device, geometric parameters, including the storage region length (*L*_st_), storage region height (*H*_st_), and body thickness (*T*_body_), significantly affect memory characteristics. Therefore, we used the aforementioned three variables for optimization. For high accuracy, simulations such as the Fermi–Dirac statistical model, Shockley–Read–Hall (SRH) recombination model, nonlocal BTBT model, Auger recombination model, trap-assisted tunneling model, quantum confinement effect, and the doping-dependent and field-dependent mobility models were considered. In addition, the GB present in poly-Si was also adopted. To apply the interface trap of GB, the reported experimental results of 1T-DRAM [24] were borrowed. The device design and analysis were performed using the Sentaurus technology computer-aided design (TCAD) tool.

## 3. Results and Discussion

Figure 3 shows the transfer characteristics of the proposed 1T-DRAM cell at different temperatures of 300 K and 358 K. Note that the threshold voltage (*V*_th_) of the proposed device was 1.05 V and 0.99 V at temperatures of 300 K and 358 K, respectively. *V*_th_ was obtained at *I*_D_ = 10^−7^ A/μm. When the temperature increased from 300 K to 358 K, the off-current increased because carrier generation was accelerated by the high temperature. In addition, because the carrier density increased the recombination rate, the retention time decreased.

Figure 4a shows the transient characteristics of the proposed 1T-DRAM cell at 358 K. The sensing margin is defined as the difference between the read “1” current and read “0” current. The sensing margin of the proposed device was 15.1 μA/μm at *T* = 358 K. The operation mechanism of the 1T-DRAM consists of program, erase, read, and hold operations. The program operation uses the BTBT mechanism to generate holes on the side of gate 2 in the body area. During the erase operation, a negative bias is applied to the drain and the potential barrier disappears; thus, the holes accumulated on the side of gate 2 in the body region move to the drain region. The bias conditions for the operation of the 1T-DRAM are summarized in Table 2. Figure 4b shows the variation in the read currents for the “1” and “0” states at different temperatures of 300 K and 358 K. Conventionally, the retention time is defined as the time elapsed until the initial sensing margin reaches 50%. The retention times of the proposed 1T-DRAM were 1.48 s and 123 ms at temperatures of 300 K and 358 K, respectively. As shown in Figure 3, increasing the temperature increased the carrier density and recombination rate and decreased the retention time.

Figure 5a,b show that BTBT vertically occurs between gates 1 and 2 in the body region during the program operation. When the program operation is used as a BTBT mechanism, it demonstrates the advantage of a lower power consumption than the impact ionization mechanism. Figure 5b shows an energy band diagram when a positive voltage of 2.0 V is applied to gate 1 and a negative voltage of −1.7 is applied to gate 2. The voltages applied to gates 1 and 2 cause band bending and BTBT. Therefore, electrons in the valence band tunnel into the conduction band, and holes are created on the side of gate 2. Because the high work function of gate 2 forms a potential well, the generated holes accumulate on side of gate 2 in the body region.

Figure 6a,b show the contour map of the hole density and energy band diagram of the proposed 1T-DRAM cell in states “1” and “0”, respectively. Note that state “1” indicates that the holes generated by BTBT after the program operation accumulate in the body area. State “0” implies a state wherein the holes further generated after the erasing operation disappear. Figure 6a shows that the hole densities in the body region on the side of gate 2 corresponding to states “1” and “0” are significantly different. Figure 6b shows the difference in the hole density with an energy band diagram. Note that the additionally generated holes lower the energy barrier, similar to when a positive voltage is applied to the body region.

Figure 7a,b show the contour map of the electron density and energy band diagram of the proposed 1T-DRAM cell in the read “1” and read “0” operations. Note that a read operation is performed via the conventional MOSFET operation. Figure 7a shows that the electron density corresponding to read “1” is higher than that corresponding to read “0”, the inversion layer is formed on the side of gate 1 in the body region, and a current flows. Figure 7b shows the effect of the electron potential in the body region depending on the presence or absence of excess holes in the storage region. The generated hole acts as if a positive voltage is applied to the body region, and when read “1” is operated, the energy barrier in the body region is lowered and a high current flows. Because the current changes depending on the presence of excess holes, it is possible to distinguish between data “1” and “0”. The difference between read “1” current and read “0” current is defined as a sensing margin in 1T-DRAM.

Figure 8a shows the contour maps of the SRH recombination rate for the proposed 1T-DRAM with different *L*_st_ in the hold state “1” operation. As shown in Figure 5a, BTBT vertically occurs between gates 1 and 2 in the body region during program operation. Moreover, in Figure 5a, the circular dotted line indicates the region where BTBT is generated, and as *L*_st_ becomes longer, the area where BTBT occurs becomes smaller. Therefore, it can be seen from Figure 8a that as *L*_st_ increases, the BTBT rate decreases and the SRH recombination rate decreases because fewer holes are generated. Furthermore, when *L*_st_ is 70 nm, SRH recombination transforms into SRH generation at the source–channel and channel–drain junctions, and in Figure 8b, the current corresponding to read “1” increases with the hold time and becomes unstable. Therefore, for all the optimization processes considered in this study, the point with the longest retention time under the condition that the current corresponding to read “1” does not increase is considered as the optimization point. In other words, SRH generation does not occur at the source–channel and channel–drain junctions. Moreover, because the read “1” and “0” current rapidly increases with time by SRH generation, the retention time is rapidly reduced at *L*_st_ = 80 nm. In Figure 8c, as *L*_st_ becomes longer, the region where BTBT occurs becomes smaller, indicating that fewer holes are generated, and the sensing margin decreases. Furthermore, it is shown that the SRH recombination rate decreases, and the retention time increases as fewer holes are generated. When *L*_st_ is 60 nm, the current corresponding to read “1” does not increase with the hold time, and the retention time is the highest; therefore, this value was selected as the optimization point. The proposed device with an *L*_st_ of 60 nm obtained a sensing margin of 4.25 μA/μm and a retention time of 410 ms at *T* = 358 K.

Figure 9a shows the contour maps of the SRH recombination rate for the proposed 1T-DRAM with different *H*_st_ values in the hold state “1” operation. As mentioned earlier, during program operation, BTBT occurs in the circular dotted line area in Figure 5a. However, a small amount of BTBT also occurs between the fin-shaped gates 1 and 2. As *H*_st_ increases, less tunneling occurs because the distance between gates 1 and 2 increases in the fin-shaped region. Therefore, as shown in Figure 9a, as *H*_st_ increases, the BTBT rate decreases, and the number of generated holes decreases, so the SRH recombination rate decreases. Figure 9b shows the current corresponding to read “1”, and the number of holes created depending on the increase of *H*_st_ decreases and the current decreases. In the end, similar to the above-mentioned *L*_st_ optimization process, when *H*_st_ is 30 nm, the current corresponding to read “1” increases with time due to SRH generation and becomes unstable. Figure 9c shows the sensing margin and retention time of the proposed 1T-DRAM cell as a function of *H*_st_. As *H*_st_ increases, fewer holes are generated, and sensing margin gradually decreases. Conversely, the retention time increases because the stored holes can be retained longer by the reduced SRH recombination rate. Notably, the proposed device with an *H*_st_ of 25 nm obtained a sensing margin of 2.51 μA/μm and a retention time of 598 ms at *T* = 358 K.

Figure 10a shows the contour maps of the SRH recombination rate for the proposed 1T-DRAM with different *T*_body_ values in the hold state “1” operation. Since BTBT vertically occurs between gates 1 and 2 in the body region, as the *T*_body_ becomes thicker, less BTBT occurs and fewer holes are created, so the SRH rate decreases. Conversely, as shown in Figure 10a, the thinner the *T*_body_, the more BTBT is generated, and the more holes are created, so the SRH rate is high throughout the body region. Figure 10b shows read “1” current of the proposed 1T-DRAM with different *T*_body_. Similar to the above-mentioned *L*_st_ optimization process, when *T*_body_ is 11 nm, the current corresponding to read “1” increases with time due to SRH generation and becomes unstable. Figure 10c shows the sensing margin and retention time for the proposed 1T-DRAM cell as functions of *T*_body_. Since electrons are accumulated in the inversion channel on the side of gate 1 and holes are accumulated on the side of gate 2, when *T*_body_ becomes thinner than a certain thickness, electrons and holes meet and recombination occurs, reducing sensing margin. In addition, when *T*_body_ becomes thicker than a specific thickness, the sensing margin decreases because the BTBT rate decreases in the program operation. For our device, the final optimized parameters were *L*_st_ = 60 nm, *H*_st_ = 25 nm, and *T*_body_ = 10 nm. Table 3 summarizes previously reported sensing margin and retention times for various 1T-DRAM cells. As can be inferred, the 1T-DRAM cell proposed in this paper exhibits excellent memory characteristics at *T* = 358 K compared with other device.

## 4. Conclusions

In this study, a novel 1T-DRAM based on a poly-Si dual-gate MOSFET with a fin-shaped structure was optimized and analyzed using a TCAD simulation. In the proposed device, a negative bias is applied to gate 2 during the hold operation to prevent the holes generated in the program operation from escaping. In addition, by using a fin-type structure, the length of the gate 2 side is geographically increased to expand the area where the hole is stored. Therefore, the proposed 1T-DRAM has improved memory characteristics as a result of which the hole storage area is expanded compared with the planar dual-gate MOSFET-based 1T-DRAM of the same channel length. The storage region length, storage region height, and body thickness have a great influence on the memory characteristics because they affect the number of holes created by BTBT during the program operation. Therefore, the optimized parameters were *L*_st_ = 60 nm, *H*_st_ = 25 nm, and *T*_body_ = 10 nm, and a high retention time of 598 ms was obtained at *T* = 358 K. In conclusion, the proposed novel 1T-DRAM demonstrates potential to replace conventional 1T-1C DRAM because it possesses a high retention time and can increase the chip density.

## Figures and Tables

**Figure 1 nanomaterials-12-03526-f001:**
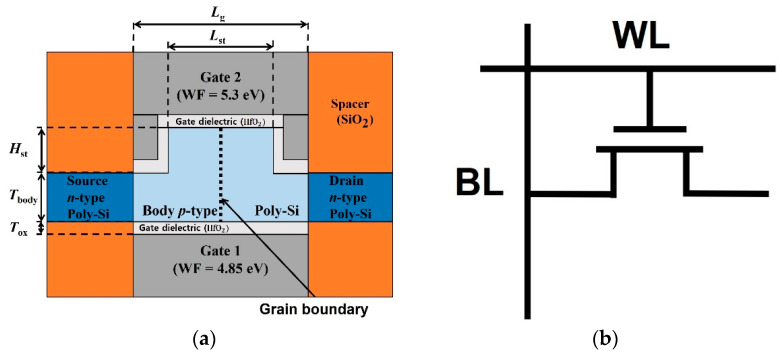
(**a**) Cross-sectional view of the proposed poly-Si dual-gate MOSFET based 1T-DRAM with a fin-shaped structure. (**b**) Simplified circuit diagram of 1T-DRAM.

**Figure 2 nanomaterials-12-03526-f002:**
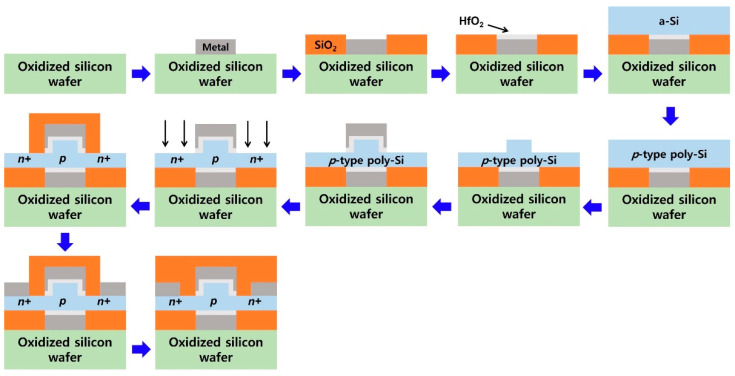
Key fabrication steps of proposed 1T-DRAM based on a poly-Si dual-gate MOSFET with a fin-shaped structure crystallized via excimer laser crystallization [23].

**Figure 3 nanomaterials-12-03526-f003:**
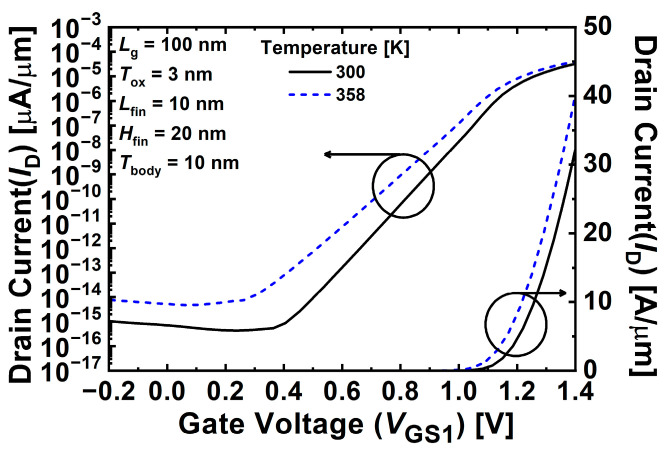
Transfer characteristics of proposed 1T-DRAM cell at different temperatures of 300 K and 358 K.

**Figure 4 nanomaterials-12-03526-f004:**
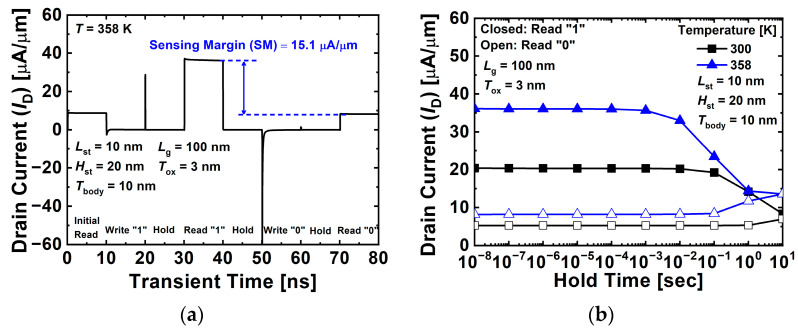
(**a**) Transient characteristics of proposed 1T-DRAM cell. (**b**) Variation of read currents in “1” and “0” states at different temperatures of 300 K and 358 K.

**Figure 5 nanomaterials-12-03526-f005:**
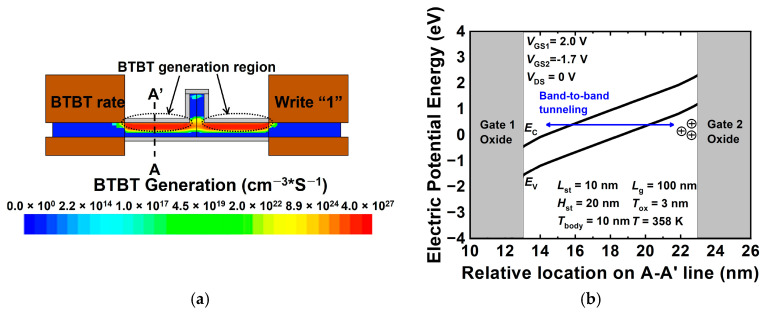
(**a**) Contour map of BTBT rate and (**b**) energy band diagram of the proposed 1T-DRAM cell in program operation (energy band is extracted 10 nm away from source–channel junction).

**Figure 6 nanomaterials-12-03526-f006:**
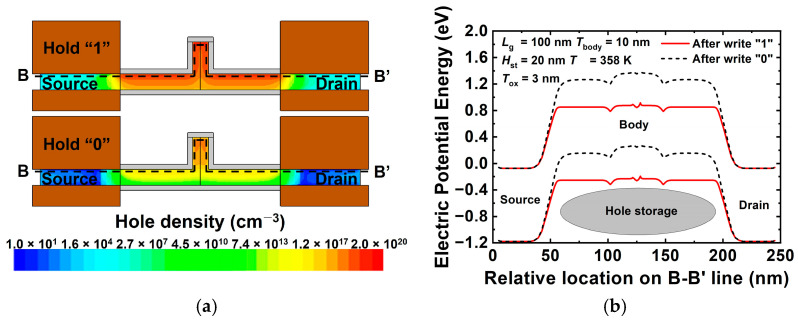
(**a**) Contour map of hole density and (**b**) energy band diagram of proposed 1T-DRAM cell in hold states “1” and “0” (energy band is extracted at 2 nm above gate 2 oxide).

**Figure 7 nanomaterials-12-03526-f007:**
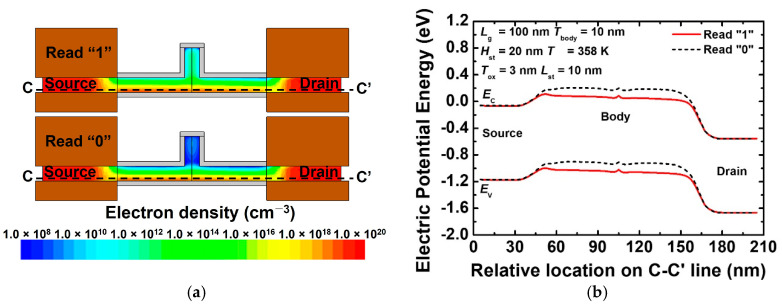
(**a**) Contour map of electron density and (**b**) energy band diagram of proposed 1T-DRAM cell in read states “1” and “0” (energy band is extracted at 3 nm above gate 1 oxide).

**Figure 8 nanomaterials-12-03526-f008:**
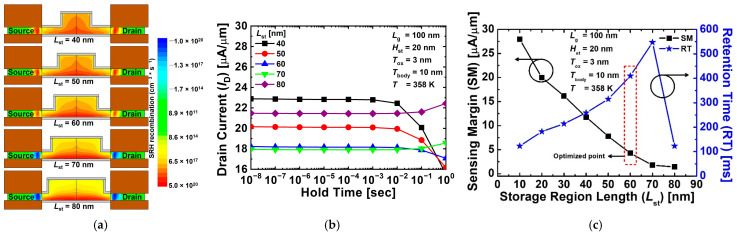
(**a**) Contour map of SRH recombination rate for proposed 1T-DRAM with different *L*_st_ in hold state “1” operation. (**b**) Read “1” current of proposed 1T-DRAM with different *L*_st_. (**c**) Sensing margin and retention time of proposed 1T-DRAM cell as a function of *L*_st_.

**Figure 9 nanomaterials-12-03526-f009:**
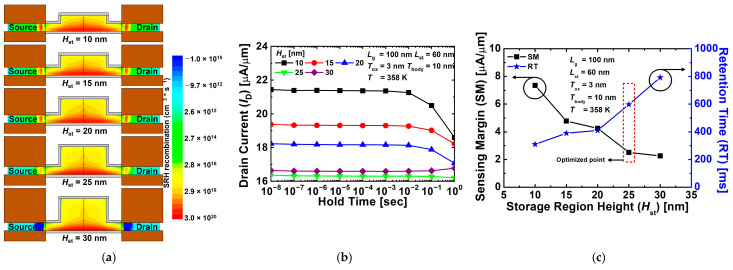
(**a**) Contour map of SRH recombination rate for proposed 1T-DRAM with different *H*_st_ in hold state “1” operation. (**b**) Read “1” current of proposed 1T-DRAM with different *H*_st_. (**c**) Sensing margin and retention time of proposed 1T-DRAM cell as a function of *H*_st_.

**Figure 10 nanomaterials-12-03526-f010:**
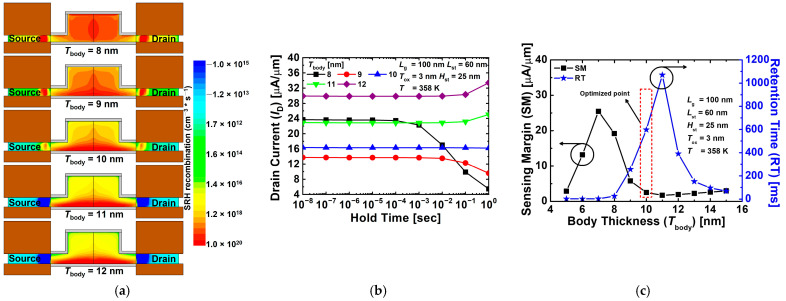
(**a**) Contour map of SRH recombination rate for proposed 1T-DRAM with different *T*_ch_ in hold state “1” operation. (**b**) Read “1” current of proposed 1T-DRAM with different *T*_ch_. (**c**) Sensing margin and retention time of the proposed 1T-DRAM cell as a function of *T*_ch_.

**Table 1 nanomaterials-12-03526-t001:** Device parameters of proposed 1T-DRAM used for simulation.

Parameter	Values
Gate length (*L*_g_)	100 nm
Body thickness (*T*_body_)	5–20 nm
Storage region length (*L*_st_)	10–80 nm
Storage region height (*H*_st_)	10–30 nm
Gate dielectric (HfO_2_) thickness (*T*_ox_)	3 nm
Source and Drain doping concentration	*n*-type, 1 × 10^20^ cm^−3^
Body doping concentration	*p*-type, 1 × 10^18^ cm^−3^
Gate 1 work-function (WF_G1_)	4.85 eV
Gate 2 work-function (WF_G2_)	5.3 eV

**Table 2 nanomaterials-12-03526-t002:** Operating bias scheme for memory performance.

	Write “1” (Program)	Write “0” (Erase)	Read	Hold
Gate 1 voltage (*V*_GS1_)	2.0 V	0.0 V	1.2 V	0.0 V
Gate 2 voltage (*V*_GS2_)	−1.7 V	0.5 V	0.0 V	−0.2 V
Drain voltage (*V*_DS_)	0.0 V	−0.5 V	0.5 V	0.0 V

**Table 3 nanomaterials-12-03526-t003:** Memory performance of various 1T-DRAM-related papers.

No	Reference	Sensing Margin [μA/μm]	Retention Time [ms]
1	[25]	5.4	68
2	[26]	28.7	79
3	[27]	0.15	320
4	[28]	52.3	11.2
5	[29]	11.7	64.2
6	[30]	6.16	131
7	[31]	6.58	340.1
8	This work	2.51	598

## Data Availability

Not applicable.

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
