# Peer review of "Design of a Capacitorless DRAM Based on a Polycrystalline-Silicon Dual-Gate MOSFET with a Fin-Shaped Structure"

_nanomaterials, 2022, doi:10.3390/nano12193526_

Round 1

Reviewer 1 Report (Previous Reviewer 1)

I recommend this paper can be accepted for publication in its current form.

Author Response

Reviewer 2 Report (New Reviewer)

Authors propose one 1T-DRAM cell based on a poly-Si dual-gate MOSFET with a fin-shaped structure. The use of this cell avoids the need of the capacitor element per cell allowing, thus, higher integration levels than classical solutions.

The paper is clear and well presented although some typos should be corrected.

As the strong point of the work, the performance of the cell is better compared to recent published papers as the sensing margin and the retention time of the proposal are better than these state of the art cells.

The main weak point of the work is that only simulation results have been used in the paper.

Author Response

Reviewer 3 Report (New Reviewer)

  • The paper overall is written and organized well

  • Can the authors comment on the metal contacts used for source and drain regions in figure 2 - suggest including these in the process flow if possible

  • Table 3 - can the authors comment on differences in retention time and sensing margin of DRAM using capacitors and 1T-DRAM?

  • Do the authors plan to perform model to hardware correlations? Suggest adding any measured hardware data if available

  • Suggest the authors include a circuit diagram representation of their proposed device

  • Figure 4a. Y-axis label says A/um, is this correct? Should it be uA/um?

  • Figure 7b - please comment on the asymmetric electric potential between the source and drain region

Author Response

Reviewer 4 Report (New Reviewer)

1.1.    The abstract is simply written. Therefore, Important finding should be mentioned in the abstract.

2.    Complete the abstract with possible applications.

3.    The authors should formulate very briefly in the introduction the novelty of the work with respect to the other works in the field.

4.    How the present author's results differ from the earlier reported ones? Give a comparison of the results and discussion part.

5. The conclusion is just a list of observations like for a technical report. Please add the science including for cause of the effects in order to advance the understanding of the studied phenomena compared to what is already known or could be expected from literature.

Author Response

This manuscript is a resubmission of an earlier submission. The following is a list of the peer review reports and author responses from that submission.

Round 1

Reviewer 1 Report

In this manuscript (nanomaterials-1865549), a capacitorless one-transistor dynamic random-access memory (1T-DRAM) cell based on a polycrystalline silicon dual-gate metal-oxide-semiconductor field-effect transistor with a fin-shaped structure was optimized and analyzed using technology computer-aided design simulation. The proposed 1T- DRAM demonstrated improved memory characteristics owing to the adoption of the fin-shaped structure on the side of gate 2. This was because the holes generated during the program operation were collected on the side of gate 2, allowing an expansion of the area where the holes were stored using the fin-shaped structure. The proposed 1T-DRAM cell exhibited a sensing margin of 2.51 μA/μm and retention time of 598 ms at T = 358 K. By reading this paper, it can be found some meaningful innovations. The data obtained by the authors well support the conclusions of the paper, and the paper is well written. Therefore, I recommend that this paper can be accepted for publication after revision.

(1) I suggest that authors can highlight the importance of studying DRAM by comparing different RAMs in the Introduction section by citing relevant reports (such as: DOI: 10.1002/aelm.202101127; 10.1039/D1NH00292A).

(2) The discussion of Figures 7, 8, and 9 is somewhat lacking, and the authors should add some detail to the description and discussion.

(3) Some grammatical errors need to be corrected by the author during the revision process.

Reviewer 2 Report

See enclosed file.
